# System Identification with Biophysical Constraints: A Circuit Model of the Inner Retina

**Cornelius Schröder**\*
University of Tübingen
cornelius.schroeder@uni-tuebingen.de

**David Klindt**\*
University of Tübingen
klindt.david@gmail.com

**Sarah Strauss**
University of Tübingen
sarah.strauss@uni-tuebingen.de

**Katrin Franke**
University of Tübingen
katrin.franke@cin.uni-tuebingen.de

**Matthias Bethge**
University of Tübingen
matthias@bethgelab.org

**Thomas Euler**
University of Tübingen
thomas.euler@cin.uni-tuebingen.de

**Philipp Berens**
University of Tübingen
philipp.berens@uni-tuebingen.de

## Abstract

Visual processing in the retina has been studied in great detail at all levels such that a comprehensive picture of the retina's cell types and the many neural circuits they form is emerging. However, the currently best performing models of retinal function are black-box CNN models which are agnostic to such biological knowledge. In particular, these models typically neglect the role of the many inhibitory circuits involving amacrine cells and the biophysical mechanisms underlying synaptic release. Here, we present a computational model of temporal processing in the inner retina, including inhibitory feedback circuits and realistic synaptic release mechanisms. Fit to the responses of bipolar cells, the model generalized well to new stimuli including natural movie sequences, performing on par with or better than a benchmark black-box model. In pharmacology experiments, the model replicated *in silico* the effect of blocking specific amacrine cell populations with high fidelity, indicating that it had learned key circuit functions. Also, more in depth comparisons showed that connectivity patterns learned by the model were well matched to connectivity patterns extracted from connectomics data. Thus, our model provides a biologically interpretable data-driven account of temporal processing in the inner retina, filling the gap between purely black-box and detailed biophysical modeling.

## 1 Introduction

In the retina, light is transduced by the photoreceptors (PRs) and processed in two layers of neuropil before the signal is sent to the brain (Fig. 1). In the outer plexiform layer, the output of PRs is shaped by feedback of horizontal cells before it is passed to the bipolar cells (BCs). In mice, 14 types of BC then relay the signal to the second synaptic layer, the inner plexiform layer (IPL). Here, the signal is

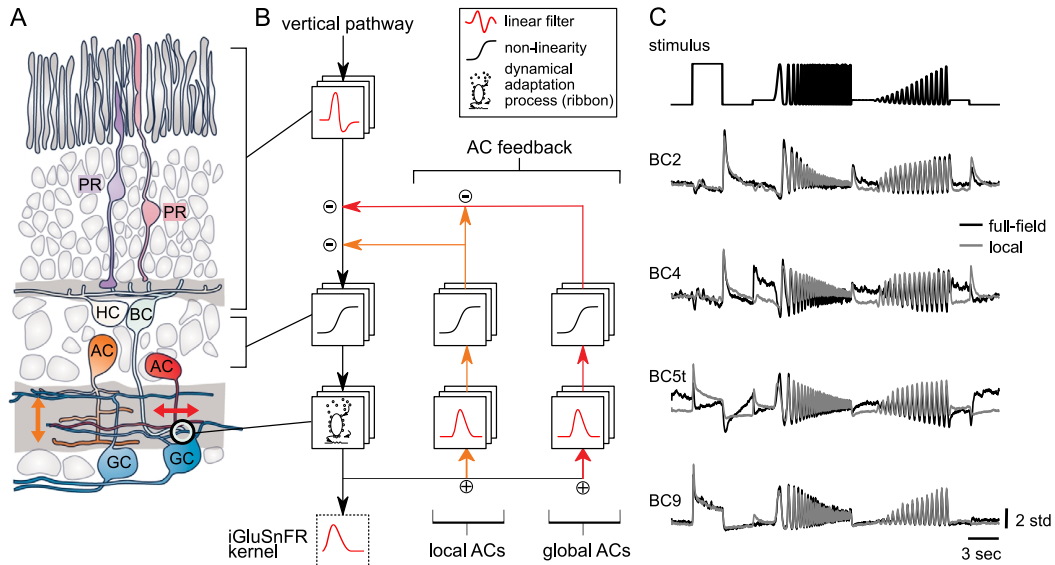

Figure 1: **Retinal circuit schematics. A.** The signalling pathways in the retina: photoreceptors (PRs) transmit the light signal to bipolar cells (BCs) before it is passed on to ganglion cells (GCs), which project the signal to the brain. In between, the signal is shaped by two classes of mostly inhibitory interneurons: horizontal cells (HCs) and amacrine cells (ACs). Figure adapted from [13]. **B.** Translation of the retinal circuit to a computational network model. We modeled two main groups of ACs: more globally and more locally acting feedback neurons. **C.** Light stimulus-evoked responses of four different types of BCs (two Off and two On cells); overlaid are traces for spatially extended (full-field) and localized (local) chirp stimuli. For details, see [5].

shaped by a complex network of more than 45 different types of mostly inhibitory amacrine cells (ACs) [1, 2, 3] and passed on to the retinal ganglion cells (RGCs), which in turn transmit the signals to visual areas in the brain. Already at the level of BCs, different parallel pathways emerge, which are tuned for specific features of the stimulus. While the different BC types have been well studied at the morphological, genetic and functional level [4, 5, 6], a comprehensive understanding of how the diverse AC types shape BC output is still missing and so far, only highly specialized AC circuits have been studied extensively [7, 2].

At the same time, the currently best-performing systems identification models for retinal neurons only account for feedforward drive and typically neglect ACs, even when mimicking the non-linear subunit structure of retinal processing [8, 9, 10, 11]. Therefore, our knowledge about the computational role of ACs is largely limited to basic principles: for example, GABA-releasing wide-field ACs mostly provide lateral feedback, while glycinergic small-field ACs predominantly mediate vertical feedback across different strata of the IPL and modulate wide-field AC input to BCs [2]. In addition, system identification models typically neglect well-understood biophysical mechanisms involved in temporal adaptation and therefore lack a clear link to the underlying biology. For example, the specialized ribbon synapse, a feature of PRs and BCs, is known to dramatically shape the temporal structure of the transmitted signal [12].

Here, we build on recent work modeling stimulus-response relationships of individual neurons extending simple linear-nonlinear models to a full-scale network model of the IPL while keeping a much higher degree of biophysical realism. Our contributions are:

1. We show how to train a network model of the IPL including ACs and a high degree of biophysical realism end-to-end to reproduce the temporal responses of all 14 mouse BC types on artificial stimuli (Figure 1 and 2).

2. We show that the predictive performance of this model is as good as that of deep recurrent models on artificial as well as on natural stimuli (Figure 3).

3. We perform *in silico* pharmacological modulations and show that blocking different groups of ACs has similar effects to what is observed in experiments (Figure 4).

4. We compare the connectivity between the different types of BCs and ACs in our model to connectomics data [14] and find that our model has learned the general rules of IPL connectivity from functional data (Figure 5).

5. Finally, we use the biological realistic components of the model to make predictions on biophysical properties of the ribbon synapses for individual BC types (Figure 6).

Thus, our model provides a biologically interpretable data-driven account of temporal processing in the inner retina, filling the gap between purely black-box modeling and detailed biophysical modeling.

## 2 Previous Work

Current models of neural processing in the retina broadly fall into two categories:

1. *Neural system identification* approaches [15] are designed to maximize the performance when predicting the activity of a retinal neuron or a population of neurons from the visual stimulus. Such models include statistical linear-nonlinear-Poisson models (LNP) and their generalizations incorporating feedback terms and non-linear subunits [8, 9] as well as models based on deep neural networks [11, 16, 17]. These approaches are able to predict the activity of retinal neurons with remarkable accuracy and subunits in the respective models can resemble presynaptic neurons [10, 18, 19, 20]. However, the models are often difficult to interpret in terms of actual biological mechanisms. In addition, they typically fail to model adaptive processes determining the temporal response kinetics in many retinal neurons.

2. *Mechanistic models* for retinal neurons are typically biophysically realistic models based on Hodgkin-Huxley equations. For example, such models have been used to model the activity of BCs and RGCs, and can do very well in accounting for adaptive processes, as they incorporate the underlying biophysical mechanisms [21, 22]. Such models are based on large amounts of biological detail and knowledge, thus enabling a mechanistic investigation into a specific computation, but are time-consuming to simulate and notoriously hard to fit to data.

To strike a middle ground between these two extremes, system identification models have been combined with an additional kinetic block that allows them to account for rapid release adaptation at synaptic sites as well as other adaptive processes [23, 24]. Additionally, a single inhibitory pathway has been incorporated in such models to account for the aggregate feedback from all ACs [25]. Here, we advance such hybrid models and combine an interpretable linear-nonlinear-release (LNR) model with a kinetic block [24] with the rich feedback structure of the whole AC network. Using this network to model temporal processing in the IPL, we obtain accurate predictions of BC activity across a wide range of stimuli while maintaining a high degree of biological interpretability. In contrast to the often involved inference necessary for biophysically realistic models [22, 26], our model is completely differentiable and can be efficiently learned end-to-end with modern deep learning frameworks.

## 3 Model

### 3.1 The Bipolar Cell Network model

Our BC network (BCN) model consists of two main parts: i) a vertical model of the 14 parallel BC channels, and ii) a model of the AC feedback (Fig. 1B). The feedback consists of a *local* and *global* pathway, activated by stimulation of center and surround component of BC receptive fields, respectively. As data, we used light-evoked responses of BCs recorded with the genetically encoded glutamate sensor iGluSnFr [5] (c.f. Appendix B). Therefore, we convolved our model output with the iGluSnFR kernel as a final step, which allows for a direct comparison to the functional recordings of BCs. In total, the model has 1,932 free parameters.

### 3.1.1 Model of the vertical pathway

The vertical pathway consists of a linear biphasic kernel, a sigmoidal non-linearity and a model of the release machinery at a ribbon synapse. The linear stage accounts for the approximately linear

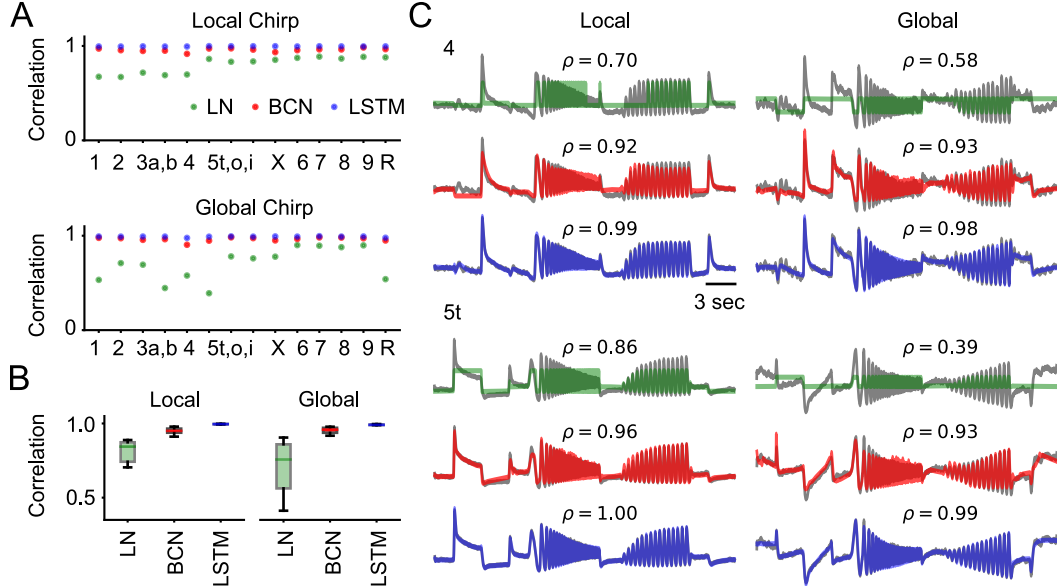

Figure 2: **Training Performance. A.** Linear correlation of the output of the three models for all 14 BC types on the local (top) and the global (bottom) chirp stimulus. **B.** Summary of the correlation across all types of BCs for the three models and two stimulus conditions. **C.** Model predictions for the Off (type 4) and On (type 5t) BC from Fig. 1C. Cluster mean traces are shown in grey and $\rho$ gives Pearson correlation coefficients.

processing of light $l$ in PRs and dendrites of BC $i$ [27]:

$$\mathrm{BC}_i^{\mathrm{in}}(t) = \int_{\tau=0}^{T} l(t - \tau) \cdot \kappa_i^{\mathrm{PR}}(\tau) d\tau.$$

The signal is then modulated by the inhibitory local and global AC feedback ($\mathrm{fb}_i(t)$, see Section 3.1.2), which is thought to be their main mode of modulation, before it is passed through a sigmoidal non-linearity to be converted into vesicle release probability $p_i$ (see Appendix A for details):

$$p_i(t) = \sigma_i \left( \mathrm{BC}_i^{\mathrm{in}}(t) - \mathrm{fb}_{\mathrm{local},i}(t) - \mathrm{fb}_{\mathrm{global},i}(t) \right) .$$

We mimic one kind of sensitivity adaptation in the retina by allowing the offset of the non-linearity to change by shifting its operating point [25]. This allows different computations in the vertical pathway for local and full-field stimuli. We used a deterministic version of the release model described in [24] to model the BC's synapse. In this model, vesicles move between three different pools in a probabilistic fashion: At each time step, vesicles are released (into the synaptic cleft) from the ready releasable pool (RRP). The replenishment of the RRP occurs in two step: Vesicles from the cytoplasm are first moved into the intermediate pool (IP) with the rate $\mathrm{IP}_{\mathrm{refill}}$, from which they are moved to the RRP in the second step (with the rate $\mathrm{RRP}_{\mathrm{refill}}$), making them available for release. To make the model deterministic, we replaced all random variables by their expected value given the present state of the different pools. This results in three simplified equations for vesicle movement:

$$\mathrm{release} = p(t) \cdot \mathrm{RRP}, \qquad \mathrm{RRP}_{\mathrm{refill}} = k_1 \cdot \mathrm{IP}, \qquad \mathrm{IP}_{\mathrm{refill}} = k_2,$$

where $k_1$ and $k_2$ are constant over time. Additionally, maximal pool sizes were learned for both the IP and RRP. For better optimization, the occupancies of the pools were sent through a sigmoidal non-linearity in each time step and thus smoothly clamped at the maximal values.

### 3.1.2 Feedback model

The feedback structure is implemented by a network of ACs (Fig. 1B). Each AC is modeled by a LN model, which receives input from all BC types with learned weights $W^{\mathrm{BC\,AC}}$. The LN part consists of a double-exponential kernel $\kappa^{\mathrm{AC}}$ (see Appendix A for details) and a sigmoidal non-linearity $\sigma$

afterwards:

$$\mathrm{AC}_i^{\mathrm{out}}(t) = \sigma_i \left( \int_{\tau=0}^{T} \left( \sum_{j=1}^{14} w_{ij}^{\mathrm{BC\,AC}} \cdot \mathrm{BC}_j^{\mathrm{out}}(t-\tau) \right) \cdot \kappa_i^{\mathrm{AC}}(\tau) d\tau \right).$$

We modeled both the local and global AC pathways, consisting of mostly glycinergic and mostly GABAergic ACs, respectively. While both groups provide direct feedback to most BCs, local ACs also act as a gate keeper for global ACs by modulating their output to BCs in an inhibitory manner (cf. Figure 1B, [13]). The two AC groups further differ in their spatial tuning. Local ACs integrate over small spatial regions (up to $300\mu m$ [2, 13]), whereas global ACs are better activated by larger stimuli, complementing their smaller counterparts. Consequently in the model, global feedback is only activated during full-field stimuli, whereas local feedback is present for all stimuli. We can thus describe the feedback for each BC $i$ in the following way:

$$\mathrm{fb}_{\mathrm{local},i}(t) = \sum_j w_{ij}^{\mathrm{AC_{local}BC}} \cdot \mathrm{AC}_{\mathrm{local},j}^{\mathrm{out}}(t) \qquad \text{and}$$

$$\mathrm{fb}_{\mathrm{global},i}(t) = \mathbb{1}_{\mathrm{global}} \cdot \left( \sum_j w_{ij}^{\mathrm{AC_{global}BC}} \cdot \mathrm{AC}_{\mathrm{global},j}^{\mathrm{out}}(t) - \sum_j w_{ij}^{\mathrm{AC_{local}AC_{global}}} \cdot \mathrm{AC}_{\mathrm{local},j}^{\mathrm{out}}(t) \right).$$

To be able to compare the learned connectivity structure, which is represented by the different weight matrices $W$, with the connectivity structure found in electronmicroscopy data, we took the number of ACs from [14] (45 ACs in total) and matched the ratio of local to global ACs to the ones identified in [3] (10:35) (see Appendix F for details).

## 3.2 Benchmark models

**LN model** As a lower bound for performance, we used a linear-nonlinear model (LN). It consists only of the two first stages of the LNR model. It has the same parameterization but does not incorporate any feedback. Its parameters were optimized using the same training schedule (c.f. Appendix C) as for the BCN model.

**LSTM** As an upper bound for performance, we used a standard long-short-term-memory (LSTM) model (implemented in pytorch with `torch.LSTMModel`) with 18 hidden dimensions, and a linear readout layer with 28 output dimensions for the 14 local and 14 global traces, respectively. The number of hidden dimensions was chosen such that the number of parameters (2,044) approximately matched the number of parameters of the BCN. The LSTM was trained on full length chirp traces. See Appendix C for details.

## 3.3 Training

The models were trained on the mean traces of the 14 BC types in response to the local/global chirp stimulus (Fig. 1C). The data was recorded in the IPL using two-photon imaging of BC output with the genetically encoded fluorescent glutamate sensor iGluSnFr and clustered into functional types using an anatomy-guided clustering approach [5] (Appendix B). The training objective was to maximize the correlation between model predictions and recorded responses across all cell types and stimuli (local and global chirp). Letting $y_{i,s}, \hat{y}_{i,s}$ denote the (mean-centered) recorded and predicted response of

Table 1: **Training and generalization performance.** All numbers indicate (rounded) mean Pearson's correlation, standard deviations are written in brackets (minimum clipped to 0.01). Best model in each column indicated in bold.

| Model | Training | | Generalization | | | |
| | Local Chirp | Global Chirp | Natural | Vary Sine | Small Sine | Large Sine |
| --- | --- | --- | --- | --- | --- | --- |
| LN | 0.80 (0.09) | 0.70 (0.17) | 0.18 (0.06) | 0.06 (0.05) | 0.02 (0.02) | 0.02 (0.02) |
| LSTM | **1.00** (0.01) | **0.99** (0.01) | **0.24** (0.06) | 0.09 (0.07) | **0.03** (0.03) | 0.03 (0.03) |
| BCN | 0.96 (0.02) | 0.96 (0.02) | **0.24** (0.06) | **0.10** (0.08) | **0.03** (0.03) | **0.04** (0.03) |

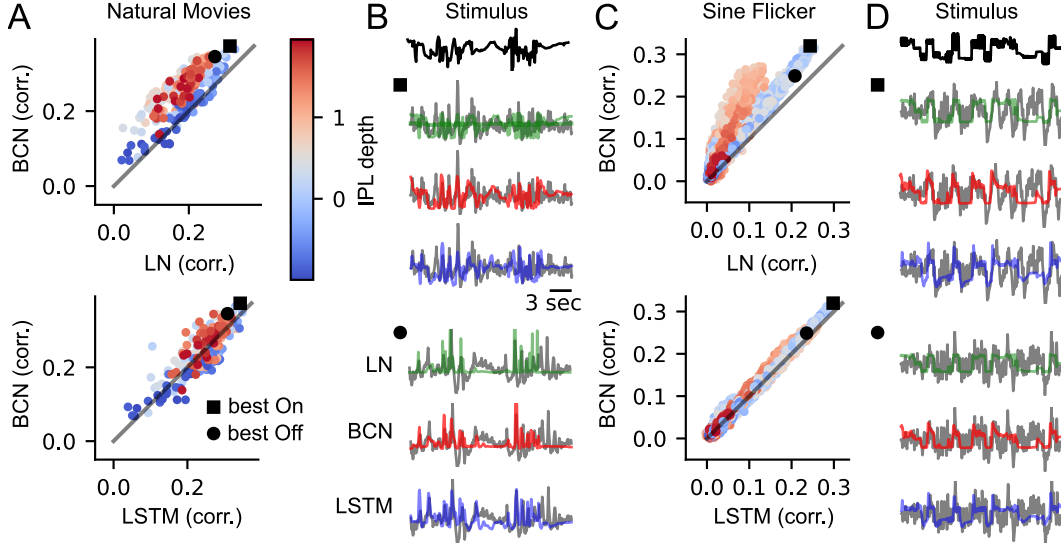

Figure 3: **Generalization Performance. A.** Each point corresponds to the Pearson correlation between recorded response and model response of a single BC terminal for the LN model (x axis) and BCN model (y axis). Color indicates IPL depth (center at $0.5$), straight line the identity. IPL depth is a good proxy for On/Off separation, Off BCs are shallower (red), On cells deeper (blue). **B.** Sample predictions for the best On and Off BC terminal (highest correlation across all models) for the three models (recorded traces in grey). **C., D.** Same as A, B but for a sine flicker stimulus of varying size. Note that the IPL depth had to be estimated differently for these recordings, which might cause a difference in scaling compared to A. The dense regions of the stimulus have high frequency sinusoidal oscillations.

type $i$ to stimulus $s$, our loss function was

$$\mathcal{L}_{correlation} = -\sum_{i=1}^{14} \sum_{s\in\{local,global\}} \frac{y_{i,s}^T \hat{y}_{i,s}}{||y_{i,s}||_2 ||\hat{y}_{i,s}||_2} \ .$$

We minimized this loss function to train the LN and LSTM model.[2] For the BCN model, we encouraged sparse connections between different types of neurons as observed in real EM data [14], by additionally adding a sparse penalty, minimizing the 1-norm of all connectivity matrices $W^j$: $\mathcal{L}_{sparsity} = \sum_j ||W^j||_1$. Finally, we weighted the two terms and optimized $\mathcal{L}_{total} = \mathcal{L}_{correlation} + \beta \mathcal{L}_{sparsity}$. All models were written in PyTorch [28] and optimized using the Adam optimizer [29] (see Appendix C for details about hyper-parameter search and learning schedule).

## 4 Results

### 4.1 Model Performance

We found that the BCN model learned to predict BC chirp responses for both local and global chirps nearly perfectly when evaluated on the training data, with performance reaching almost that of the LSTM model (Fig. 2A, B; Table 1). In contrast, the LN model performed noticeably worse, failing to capture salient response features such as a slowly decaying response during the first onset of light (Fig. 2C). The BCN model was able to model this adaptative process accurately. While the LSTM model likewise captured this process and even achieved slightly higher correlation values, it showed signs of over-fitting as it predicted little noise ripples in the data (Fig. 2C, 4, left) .

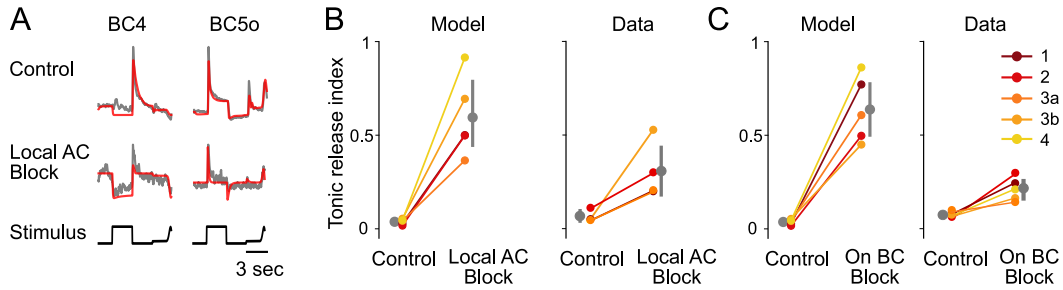

Figure 4: *In silico* **pharmacological experiments. A.** The model reproduces type-specific modulations of BC responses induced by drug application, all showing local responses (model in red, data in grey). **B.** Tonic release index (defined in Appendix E) for Off BCs under control condition and blocking of glycinergic (local) ACs (errorbars indicating bootstrapped 95% confidence intervals, Model: p=0.004 for a paired t-test). **C.** Tonic release index for Off BCs under control condition and blocking of the On BC pathway (Model: p=0.002 for a paired t-test). All figures of experimental data adapted from [5].

To probe the generalization performance of the model to unseen stimuli, we used additional recordings of BC terminals in response to natural movies and to sinusoidal flickering stimuli of constant and varying spatial size. To obtain a time series approximating the contrast statistics across the spatial receptive field of a single BC for the spatially inhomogeneous stimuli, we filtered these stimuli with a spatial difference of Gaussian kernel (see Appendix B) .

We found that the BCN performed better than the LN model and as good as or better than the LSTM on the hold out data (Fig. 3; Table 1). In particular, we found a more pronounced performance gain for Off compared to On BCs. We want to highlight that the training data were averaged over many animals/ROIs/repetitions, while the natural movie dataset consists of averages over only five repetitions and the sine dataset of single trial traces, making the two latter substantially more noisy. Furthermore, the datasets were collected under different experimental conditions, making it a harder generalization task because of the domain shift. This resulted in lower correlation levels for the generalization data compared to the training data. Additional data sets with variations of the sine stimuli are shown in Appendix D.

## 4.2  *In silico* **Pharmacological Manipulations**

We tested three different pharmacological manipulations *in silico* and compared their effects to previously obtained experimental results [5]. See Appendix E for details of the implementation. Blocking of local feedback led to more transient responses in On BCs and an increased modulation of release below baseline for Off BCs, in line with experimental findings (Fig. 4A, B). This is thought to result from an increase in tonic glutamate release caused by blocking cross-over inhibition from the On pathway mediated by small-field, glycinergic ACs. To confirm this idea, we additionally *in silico* blocked On BCs, that provide the excitatory drive for cross-over inhibition, and observed similar effects consistent with experimental data (Fig. 4C). This suggests that our model learned the circuit motif of cross-over inhibition. In addition, blocking local feedback decreased the correlation of local and global chirp responses of model BCs significantly ($p = 0.036$) due to dis-inhibition of global feedback (Appendix E). While the decrease in correlation was less pronounced in the model compared to experimental BC responses, this suggests that the model learned the gating of global by local feedback. Finally, blocking global feedback instead resulted in an increase in correlation between local and global chirp responses compared to the control condition, matching experimental data (Appendix E).

## 4.3  **Connectivity Analysis**

Next, we compared the connectivity weights in our model to the connectivity of the IPL. For this, we used an EM data set consisting of all contacts between neurons of the inner retina [14] (for processing of the data set, see Appendix F). These contacts only represent potential synaptic contacts, as the data

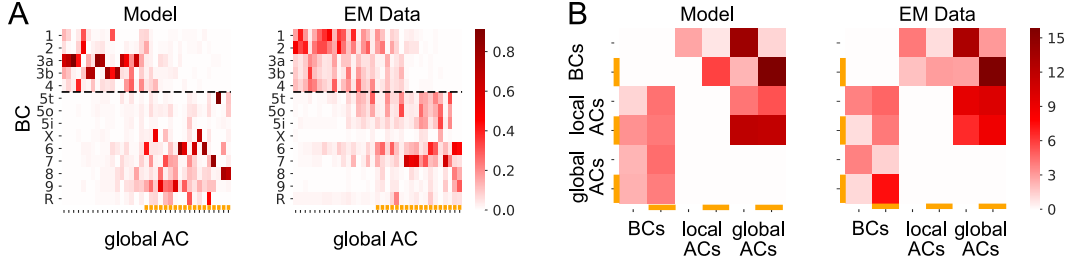

Figure 5: **Connectivity analysis. A.** Example weights of the best model and the EM data from [14] for BC to $AC_{global}$ connectivity $W^{BC\ AC_{global}}$. See Appendix F for the full weight matrices. Orange lines indicating classification as On-cell. **B.** Connectivity matrices reduced to On/Off entries. See Figure 11 for examples of randomly sampled matrices.

set does not contain any synaptic markers. For the statistical tests in Fig. 9 we compared the weights of the 20 best training runs to account for different solutions during the non-convex optimization.

In order to compare ACs between model weights and connectivity data, we ordered them according to the ratio of Off to On BC input (Fig. 5A). ACs which predominately received input from On BCs were classified as On ACs, and vice versa. The connectivity matrices revealed a block diagonal structure in both the model and the EM data (Fig. 5A; see Appendix Fig. 10 for full connectivity matrix), which was even more striking if all connections of cells with same response polarity (On/Off) were combined (Fig. 5B). This suggests that feedback within On and Off layers is much stronger than between the two layers. Overall, the learned connectivity weights were slightly sparser than in the EM data (Fig. 9A). To assess whether the similarity between model and EM connectivity matrix was due to chance, we constructed a random connectivity model. Each entry of the random connectivity matrix was randomly drawn from the EM data distribution and the complete matrix preprocessed in the same way as before (see Fig. 10 for two random example matrices). However, the correlation of the model connectivity and the EM data was significantly higher than the correlation with the random model (Fig. 9B,C, $p = 0.002$, $p = 0.018$ for the best model respectively). We also found that the fraction of On ACs among global ACs (Fig. 9D) and the ratio of On to Off AC input to the BCs (Fig. 9 E) matched the EM data well. In contrast, the fraction of On ACs among local ACs was more comparable to the random model, suggesting that this feature of IPL connectivity could not be learned from the current limited functional data.

## 4.4 Model-based Prediction of Biophysical Properties

Further, we show how our interpretable BCN model can be used to make predictions about cellular properties on the biophysical level. Inspecting the model parameters for the synaptic release of BCs, we found that RRP capacity was correlated with the global transience index (a measure of the activity decay after a large activation, computed on the experimental data, for details, see [5]; Pearson, $\rho = 0.306$ bootstrapped 95% confidence intervals $[0.204, 0.395]$, 20 best models, Fig. 6A). This is surprising, since we would have expected to find smaller RRP capacities in conjunction with more transient cell types. Interestingly, the RRP capacity together with the transfer rate from IP to RRP divided the BC types into clearly distinguishable clus-

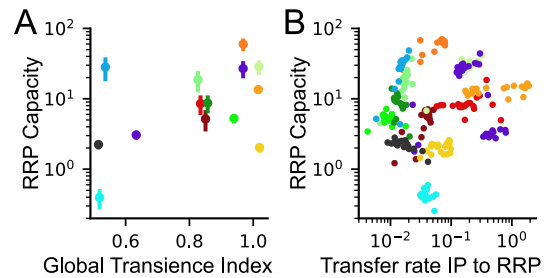

Figure 6: **Model-based Predictions. A.** Global transience index (as in [5]) and RRP capacity of the ribbon block. **B.** Model parameters of the ribbon block by BC type for the best 20 models. Red/yellow Off BCs, blue/green On BCs. See Appendix E for detailed color code.

ters (Fig. 6B), suggesting that measuring these parameters experimentally could be important for understanding the emergence of different temporal processing channels in the inner retina.

### 4.5 Ablation experiments

Finally, we investigated the contribution of the different model parts by training two ablated models, one without AC feedback terms (a deterministic version of the LNR model in [24]) and one without vesicle release block but all AC feedback structure. The models were trained in the same framework as the BCN. The ablated models all showed lower training correlation than the BCN model. In particular, they failed to capture certain features of the chirp response: For the LNR model, we found missing "feedback features" (e.g. higher baseline in some On cells or missing responses to small amplitude On steps, Fig. 12A). For the BCN without vesicle release block, we found a mismatch in the response to the long On/Off phase of the stimulus as the adaptation processes could not be fully captured (Fig. 12B). In contrast to this, we found for other cells that some of the functional properties of the release block can be approximately captured by the feedback structure (not shown).

As an additional experiment we compared the influence of training and model structure, which is in general a nontrivial task. For this, we modified the best performing BCN model. We kept all linear filters, non-linearities and the release blocks fixed but used randomly initialized feedback connectivity weights. This randomly initialized model performed poorly on the training data, but - depending on the strength of the feedback - did surprisingly well on the test data. The used evaluation procedure (we assign traces from the test data to the output channel of the model with the highest correlation), surely produces an upwards bias. Nevertheless, it seems that for some test conditions, already some unspecific feedback is sufficient, and the model structure contributes strongly to its overall performance.

## 5 Discussion

We trained a network model of temporal processing in the IPL including known structural constraints as well as biophysically inspired mechanisms to predict the functionally distinct responses of all 14 mouse BC types to different stimuli. It generalizes well and performs on par with a recurrent black-box model. Importantly, *in silico* pharmacology manipulations revealed that the model learned "cross-over inhibition" and "gating of global by local feedback" from the functional data, two of the central circuit motifs of the IPL. In addition, the connectivity structure of the model closely resembled that found in EM data. Furthermore, the model predicts that the 14 BC types can be clearly distinguished by the parameters of their synaptic release cascade, a prediction which remains to be tested. We emphasize that such predictions would not be easily possible from a pure systems identification approach. Our ablation experiments finally show, that all model parts are crucial to reproduce the detailed fingerprints of the different BC types, but also that the model design with its specific feedback structure seems to capture some circuits mechanisms even before fine tuning its weights.

Of course, our BCN model is but a first step in a comprehensive model of the IPL. On a technical level, it would be highly desirable to perform inference for BCN parameters using recent advances in Approximate Bayesian Computation [30, 31]. However, these approaches are typically limited to models with dozens of parameters [22, 24]. When the technical challenges involved have been solved, this will allow for the identification of degenerate solutions and dependencies in the parameter space. Further, the BCN neglects most forms of more involved spatial processing or processing across light levels. In both cases, different sets of ACs are recruited across different stimulus conditions [2]. Therefore, including data from such conditions may be key in further understanding in how far the connectivity structure of the IPL follows computational demands.

### Broader Impact

We present a model for temporal processing in the inner retina that combines system identification approaches with biophysically interpretable modules. The investigation of these modules allowed us to not only reproduce earlier experimental observations but also make predictions for the underlying biological system. First, this firmly grounds predictive models of neural activity in the biology of the underlying neural system, which is of high interest from a theoretical perspective. Second, the developed techniques for combining predictive and mechanistic models can likely be applied in other regions of the central nervous system, as the necessary data to provide the mechanistic constraints become available. Finally, our model may provide a first step towards establishing data driven *in*

*silico* experiments. This is not only valuable in the interest of reducing animal research, but also for assessing the mechanisms of retinal degeneration and may inform future generations of targeted therapies aimed at curing the underlying diseases. At this time, we cannot envision any negative consequences to arise of this research.

## Acknowledgments and Disclosure of Funding

This research was funded by the Deutsche Forschungsgemeinschaft through a Heisenberg Professorship (BE5601/4-1) and the CRC 1233 "Robust Vision" (grant number 276693517), the Excellence Cluster 2064 "Machine Learning — New Perspectives for Science" (grant number 390727645) and the Priority Program "Computational Connectomics" (BE5601/2-1, EU42/9-1) as well as the German Ministry of Education and Research through the Bernstein Award to PB (FKZ 01GQ1601) and the Tuebingen AI Center (FKZ 01IS18039A). Unrelated to this work, MB is an Amazon scholar and co-founder of Deepart UG, and Layer7 AI GmbH.

We thank Jonathan Oesterle and Timm Schubert for support. We also thank Lukas Schott for advise on the LSTM model. Finally, we thank the NeurIPS reviewers for taking their time to provide feedback and suggest further experiments that significantly improved the quality of this work.

## Footnotes

\*Equal contribution. Code available at `https://github.com/berenslab/bc_network`.

[2]As two-photon imaging data is on an arbitrary scale, we did not learn the final linear transformation that minimizes the squared error of our model predictions. This linear transformation can simply be computed after fitting.

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
