[Supplementary Material]

# Appendix

## A Model details

### A.1 Vertical Pathway

The linear filter $\kappa^{\text{PR}}$ in the vertical pathway was modeled as a biphasic filter [32, 33] which depends on a rise and decay constant ($\tau_r$ and $\tau_d$) and additional phase parameters ($\phi$ and $\tau_{\text{phase}}$). We used additionally one free parameter $\gamma$ as in [24], which is stretching/compressing the kernel on the time axis and inferred from the data. This leads to the kernel

$$\kappa^{\text{PR}}(t, \gamma) = \frac{-\left(\frac{t}{\gamma \tau_r}\right)^3}{1 + \frac{t}{\gamma \tau_r}} \cdot \exp\left(-\left(\frac{t}{\gamma \tau_d}\right)^2\right) \cdot \cos\left(\frac{2\pi t}{\gamma \phi} + \tau_{\text{phase}}\right) \ ,$$

which was normalized to have norm one. The remaining parameters were held constant: $\tau_r = 0.05$, $\tau_d = 0.05$, $\tau_{\text{phase}} = 100$ and $\phi = -\pi/7$.

We used a sigmoidal non-linearity $\sigma$ with two parameters for the offset and the slope ($x_0$ and $k$):

$$\sigma(x) = \frac{1}{1 + \exp(-k(x - x_0))} \ .$$

### A.2 Amacrine Cell Feedback

The synaptic integration of ACs was modelled as a double exponential kernel $\kappa_{\text{AC}}$ with two time parameters for rise and decay ($\tau_r$ and $\tau_d$) per AC:

$$\kappa^{\text{AC}}(t) = \exp\left(\frac{-t}{\tau_d}\right) - \exp\left(\frac{-t \cdot (\tau_d + \tau_r)}{\tau_d \cdot \tau_r}\right) \ .$$

As non-linearity we took the same sigmoidal non-linearity as in the vertical pathway.

## B Data

We used three different sets of data. Published recordings in response to the chirp stimulus (for details, see [5]) were used for model fitting. To test generalization performance, we used published recordings in response to sine flicker (for details, see [34]) as well as newly recorded responses to natural movies (see Section B.2). We want to highlight that the datatsets were recorded under different experimental conditions (for the chirp dataset scans in the xy plane were used, for the other datasets scans in the xz plane) and by different experimenters and are therefor inherently more difficult to predict.

### B.1 Chirp

The model was trained on the cluster mean traces of the 14 BC types recorded and clustered in [5] in response to local/full-field chirp stimulus (Fig. 1C). The chirp stimulus was slightly intensity and time corrected (as in [22]) to account for slight deviations in frame rate of the projector (not exactly at 60 Hz) and an offset between digital synchronization signal ("trigger") and stimulus presentation ($\sim$34 ms), as well as for the gamma distortion of the projector.

### B.2 Natural Movies

The natural movie consisted of 108 5-second sequences (64x64 pixels, $7\mu m$ pixel size) extracted from several Hollywood movies displayed at 30 Hz. Next to these "training" sequences, which were displayed in a randomized order to avoid adaptation artefacts, 5 5-second "test" sequences were displayed in a fixed order at 3 time points (beginning, middle and end). This allowed quantifying response consistency during the time of stimulus presentation. We recorded BC responses to these movies in n=1 Pvalb[Cre] mouse (Jackson laboratory, JAX 008069) using two-photon glutamate imaging in vertical optical slices (64x56 pixels @11.16 Hz) of the IPL, as described previously [35, 5]. All animal procedures were approved by the governmental review board (Regierungspräsidium Tübingen, Baden-Württemberg, Konrad-Adenauer-Str. 20, 72072 Tübingen, Germany) and performed according

to the laws governing animal experimentation issued by the German Government. Regions-of-interest (ROIs) were determined using local image correlation and glutamate traces of single ROIs corresponded to the output of single BC axon terminals (for details, see [5]).

To transform the videos into 1D temporal input sequences, we fitted a difference-of-Gaussians (DoG) filter of the same spatial size as the movies:

$$\Gamma_{\sigma,K\sigma}(x,y) = I * \left( \frac{1}{2\pi\sigma^2} e^{-(x^2+y^2)/(2\sigma^2)} - \frac{1}{2\pi K^2\sigma^2} e^{-(x^2+y^2)/(2K^2\sigma^2)} \right)$$

by maximizing the correlation between the output and the recorded traces.

### B.3 Sine Flicker

The sine flicker stimulus consisted of three different conditions: in the first condition the retina was stimulated with a small spot (with a diameter of $100\mu m$) with varying frequencies and intensities, in the second condition with a large spot (with a diameter of $800\mu m$) with varying frequencies and intensities and in the last condition the spot size changed additionally randomly between 10 different diameters (from 100 to $800\mu m$). The intensity varied between 10% and 100% contrast and the frequency between and 1Hz and 8Hz, all parameters were constant over trials for one second. See [34] for more details.

For the stimulus of varying size we collapse the spatial dimension with a normalized one dimensional DoG filter $\Gamma$ which was fitted to maximize the correlation between the stimulus and collapsed responses. The un-normalized filter had the form

$$\Gamma_{\sigma_1,\sigma_2,w}(x) = e^{-x^2/\sigma_1} - w \cdot e^{-x^2/\sigma_2} .$$

### C   Random Search and Training Details

All models were trained using the Adam optimizer [29] and an adaptive learning rate schedule for which we performed a random search over hyperparameters. The schedule is as follows: 1) draw an initial learning rate from Log-Uniform $(0.01, 1)$; 2) Train until the loss has not increased for $N$ steps; 3) lower the training rate by $0.5$; 4) iterate over 2) and 3) for $M$ epochs. Where $(N, M)$ are hyperparameters that are randomly selected from $(5, 100)$ and $(3, 10)$ respectively. At $1,000$ steps the optimization was stopped regardless of convergence (most top performing models converged at around $200 - 500$ steps).

For the BCN model, we added two regularization parameters to ensure biologically plausible solutions. The first was a loss on the variance of the biphasic kernel speeds. Without this, we observed speed difference of up to 3x which is not biologically plausible (the biological range being $(1 - 1.5)$ [35]). Secondly, we also minimized the variance of the mean and variance of the release output (after the ribbon block) for each BC type. Without this loss, we had observed that the different types learned drastically different scalings (due to the scaling invariance introduced by the final normalization). This made the learned weights difficult to compare and sometimes shifted the operating range for local and global responses in an artificial way. Each of these regularizers was weighted with a hyperparameter that was included in the random search.

### D   Training Results

### D.1   Results for local and global flicker stimulus

Figure 7: **Generalization performance for local and global sine flicker stimulus.** Left, same as Fig. 3 A for small sine flicker stimulus. Right, same as Fig. 3 C for large sine flicker stimulus.

## D.2 Full Training Performance

|  | Model, Data | 1 | 2 | 3a | 3b | 4 |
|---|---|---|---|---|---|---|
| | LN, Local | 0.67 | 0.67 | 0.72 | 0.69 | 0.70 |
| | LSTM, Local | 0.99 | 0.99 | 1.00 | 0.99 | 0.99 |
| Off BC | BCN, Local | 0.97 | 0.96 | 0.95 | 0.95 | 0.92 |
| | LN, Global | 0.53 | 0.71 | 0.69 | 0.45 | 0.58 |
| | LSTM, Global | 0.99 | 0.99 | 0.99 | 0.99 | 0.98 |
| | BCN, Global | 0.98 | 0.97 | 0.96 | 0.96 | 0.90 |

|  | Model, Data | 5t | 5o | 5i | X | 6 | 7 | 8 | 9 | R |
|---|---|---|---|---|---|---|---|---|---|---|
| | LN, Local | 0.67 | 0.67 | 0.72 | 0.69 | 0.70 | 0.86 | 0.83 | 0.84 | 0.85 |
| | LSTM, Local | 0.99 | 0.99 | 1.00 | 0.99 | 0.99 | 1.00 | 1.00 | 1.00 | 1.00 |
| On BC | BCN, Local | 0.97 | 0.96 | 0.95 | 0.95 | 0.92 | 0.97 | 0.97 | 0.96 | 0.93 |
| | LN, Global | 0.53 | 0.71 | 0.69 | 0.45 | 0.58 | 0.39 | 0.78 | 0.76 | 0.78 |
| | LSTM, Global | 0.99 | 0.99 | 0.99 | 0.99 | 0.98 | 0.99 | 0.99 | 0.99 | 0.99 |
| | BCN, Global | 0.98 | 0.97 | 0.96 | 0.96 | 0.90 | 0.95 | 0.98 | 0.97 | 0.95 |

## D.3 Selection of the Best Models

To identify the best the 20 models for later analysis, we took all models within a 2% performance range of the highest correlation and under these we took the ones with the largest weighted sum of penalty weights ($\beta$ and weights for the kernel speed and variance of the release and mean output (described in C)). For the best model we took the highest penalized model within a 1% performance range.

## E Details for *in silico* Pharmacological Manipulations

Figure 8: **Correlation between local and global traces in response to the chirp stimulus. A.** Blocking global feedback. Colors are coding different BC types. For data: black dot is the mean correlation of the cluster mean traces, grey is the mean correlation of the single trials. (errorbars indicating bootstrapped 95% confidence intervals, Model: p = 0.025, paired t-test). **B.** Blocking the local feedback (Model: p = 0.036, paired t-test) All data plots are adapted from [5].

We performed three different pharmacological experiments *in silico*: (1) blocking global feedback from wide-field ACs (*in vitro* using a combination of the GABA receptor blockers TPMPA and gabazine), (2) blocking local feedback from small-field ACs (*in vitro* using the glycine receptor antagonist strychnine), and (3) selective blocking of the On BC pathway (*in vitro* using L-AP4, an agonist of the mGluR6 receptors expressed by On BCs).

**Blocking of global feedback** We modified the parameters of the best model by setting $W^{AC_{global}BC}$ to zero. Therefore, we blocked the global AC feedback, but the global and local models still differed by the offset parameters of the non-linearities of the vertical pathway.

**Blocking of local feedback**   We modified the parameters of the best model by setting $W^{\mathrm{AC_{local}BC}}$ and $W^{\mathrm{AC_{local}AC_{global}}}$ to zero. Therefore we blocked the direct local feedback as well as the gating function of the local ACs for the global ACs.

**Blocking of the On pathway**   We modified the parameters of the best model by setting the entries corresponding to the On cells of $W^{\mathrm{BC\,AC_{global}}}$ and $W^{\mathrm{BC\,AC_{local}}}$ to zero. Therefore we blocked the On input to both local and global ACs, and looked then only at the responses of the Off BCs.

**Tonic Release Index**   To calculate the Tonic Release Index (TRI), we followed the definition in [5]: We took the model responses to the chirp stimulus and subtracted first the baseline for each cell (response to the stimulus step of low light intensity) to get the baseline corrected response $r$. We than calculated

$$\mathrm{TRI} = \frac{\sum_t |r_-(t)|}{\sum_t |r_+(t)| + \sum_t |r_-(t)|} \;,$$

where $r_- = r \cdot \mathbb{1}_{r<0}$ and $r_+ = r \cdot \mathbb{1}_{r>0}$ .

## F   Connectivity analysis

Figure 9: **Connectivity analysis. A.** Connectivity weight distribution after normalization for EM data and the best model. **B.** Correlation of the weight matrices between the 20 best models and the EM data/1,000 randomly sampled (and sorted) weight matrices. Red dot indicating best model. **C.** Correlation of the On/Off collapsed weight matrices (as in B) between the 20 best models and the EM data/1,000 random samples. Red dot indicating best model. **D.** Fraction of local and global ON-ACs for the EM data (black) and the mean of the 20 best models (red) and randomly sampled matrices (grey boxplot). The models did not show high variations and we omit to show the spread of the data. **E.** Histogram over Off/On input ratios for all ACs of the data and the best model.

### F.1   Preprocessing of the EM data

To compare the learned model weights with the contact areas of the EM in [14], we processed the data in the following way. Firstly, we extracted contacts involving ACs and BCs (excluding ones with a contact area above 5 $\mu m^2$ as they likely do not correspond to synapses). We summarized these contacts according to cell type, resulting in a connectivity matrix of BC and AC types. We found that three types of cells (On and Off starburst amacrine cells, rod bipolar cells) dominated the matrix. Thus, we normalized the corresponding entries in the matrix such that their mean would match the mean of the remainder of the matrix (non-zero elements). As the dominant functionalities of these cell types, motion detection and signal transmission during low light levels, are not exploited by the stimulus used in our experiments, we believe this normalization step makes the comparison to the model weights more reasonable. The ACs in the "Helmstaedter dataset" are not classified as local or global ACs. Therefore, we choose 9 small-field and 1 medium-field AC (characterized in [36]), which were also identified in the Helmstaedter dataset to be local ACs. Furthermore, we verified that they used glycine as their neurotransmitter, if possible in mouse [37, 38], and otherwise in another mammalian species [39, 40]. For 8 out of the 10 cells, we were able to confirm glycine as their neurotransmitter in this way. By this procedure, we think that we separated the identified ACs in [14] into local and global ACs as good as possible with the currently limited knowledge about AC types.

## F.2    Preprocessing of the Model Weights

Before normalizing and ordering the connectivity matrices of the model as described in the main text, we additionally adjusted the input weights to the ACs with the magnitude of the BC output. More precisely we scaled the weights of $W^{BCAC_{\text{local}}}$ and $W^{BCAC_{\text{global}}}$ which are corresponding to BC $i$ with the standard deviation of the output of BC $i$. The AC to BC connections were not scaled, since the output of the ACs are all on a same scale due to the used sigmoidal non-linearity.

## F.3    Connectivity Matrices

Figure 10: **Complete connectivity matrices. A.** Weights for the best model. **B.** Preprocessed connectivity matrix for the EM data. **C., D.** Examples of randomly generated connectivity matrices used for calculating a baseline distribution. All matrices are sorted and normalized in the same way (see Section F). Orange bars indicating classification as On cell.

Figure 11: **Randomly sampled and collapsed matrices. A., B.** Same matrices as C,D in Figure 10 but reduced to On/Off connectivity.

## G    Ablation experiments

Figure 12 shows two example traces for both ablated models which illustrate the qualitative influence of the model components. As all celltypes show different adaptation and feedback features, also the effect of ablating model components varies among the celltypes, but here we show some common principles.

Figure 12: **Predictions for ablated models on the chirp stimulus. A.** BCN without feedback: prediction for global BC1 (A.1) and global BCR (A.2) response. **B.** BCN without release block: prediction for local BC4 (B.1) and local BC6 (B.2) response. Stimulus is shown in black.