[Reviews · NeurIPS 2020]

Review 1

Summary and Contributions: This work makes significant progress towards a complete model of the inner retina. They strive to attain a balance between the statistical models and mechanistic models of neural computation. The statistical model consists of the inhibitory feedback mechanism using local and global amacrine populations, the mechanistic model consists of ribbon synapse along with the standard linear-nonlinear model of bipolar processing. After learning the parameters on light-evoked trial-averaged responses to chirp stimulus in mice bipolar cells, they validate the model using natural scenes and gratings. They compare simulated pharmacological experiments to real experiments and attain insights on the contribution of inhibition to explain these results; compare the inferred weights to EM data and give predicted clusters of synapse properties that could be tested in future experiments.

Strengths: - The authors have attempted to extensively validate their model against experimental data- various visual stimuli, EM, and pharmacology. - Combine two different kinds of circuit model components into a single model. - Generalization to different stimuli - train on a well curated visual stimulus, but test on a novel, more general stimulus class. - Attempt to test their model with various sources of experimental data. - Potentially relevant for neuroscientists who attend NeurIPS.

Weaknesses: If the main focus of the paper is to combine amacrine based feedback with ribbon synapse, the authors should emphasize the advantages of combining these two components. -- Are there any unique response properties that arise due to the interaction between ribbon synapses and amacrine feedback ? -- What is the contribution of ribbon synapse to the model validation results presented in the paper? It seems that all of the validation against experimental data is dependent on modeling the amacrine pathway. Do results in Section 4.1, 4.2, 4.3 change after replacing the ribbon synapse with a more commonly used linear operation? -- Similarly, how does the clustering in Figure 6 change if you do not include the Amacrine network ? Lack of spatial stimulus, single light level and use of only trial-averaged data. The usage of trial-averaged data makes the stochastic ribbon model considerably simpler. The training method might not generalize to single-trial data. The inferred parameters might not be usable in the stochastic/ single trial setting .. Other factors left out: multiple recordings. Are the scientific claims still true when accounting for variations across experimental preparation? Analysis on one another dataset might be good.

Correctness: The analysis in Section 4.3 and 4.4 rely on comparing inferred weights. Does the inference procedure guarantee a unique solution? -- Is the inference convex? -- If the learning does not guarantee unique weights, how would the results change when you use weights that are estimated using different runs of training? How are the results of the simulated pharmacology and connectivity analysis affected by the assumptions (constraints) of the model, and by learning model parameters from data? -- Since the amacrine feedback is constrained to be inhibitory, removing this feedback will trivially lead to an increase in firing rate. This could potentially explain the results in Figures 4B, C. -- One way to isolate the contributions of modeling assumption might be to run the analyses on the model before its parameters are learned from the experimental data. -- An alternate approach might be to allow the model to learn either inhibitory or excitatory weights, and if the learned weights are negative, then it suggests the necessity of inhibitory connections. -- Similarly, would block diagonal structure in Figure 5A arise if you take a random connectivity matrix and just sort the cells based on the ratio of OFF:ON inputs? -- In Figure 5B, many of the connections are not possible (white region) due to the model architecture.

Clarity: The paper is generally well written. Since simplifications on biophysically detailed modeling is the focus of this paper, then a description of why the original problem is hard and what led to simplifications might be useful to add. The section on synaptic release modeling is unclear (lines 104-115). RRPrefill, IPrefill are undefined. Figure numbering is inconsistent in some places. For example, Figure 9 in lines 207-215 refers to Figure 3 in Supplement. Figure 3A shows that EM is sparser than the model, but the statement in line 206 says otherwise. Global Transience Index must be defined in the main text. Include equation numbers. Why is the performance for OFF cell types higher compared to ON (Figure 3, line 175). Is this due to some aspect of training data, or is this a new scientific finding?

Relation to Prior Work: The specific model presented in this paper is novel, according to me. Previous attempts to fit biophysically realistic models using statistical methods must be included in previous works. (For example, see this paper: http://papers.nips.cc/paper/2394-maximum-likelihood-estimation-of-a-stochastic-integrate-and-fire-neural-model.pdf). There are many such works, and this literature must be referenced in section starting Line 70. The paper compares to two previous models - LNP model and an LSTM model. While there is good reason to compare to these models (commonly used baseline and state-of-the-art respectively), there are other models that also account for some mechanisms included in this paper. For example the GLM model from Pillow et al. 2008; BCM feedback model from (Real, Esteban, et al. "Neural circuit inference from function to structure." Current Biology 27.2 (2017): 189-198.); or the model of ribbon synapse without the feedback term (Schröder, Cornelius, et al. "Approximate bayesian inference for a mechanistic model of vesicle release at a ribbon synapse." Advances in Neural Information Processing Systems. 2019.). Description of differences from those model might be useful. A comparison of model performance to these models will help in understanding what the joint model of feedback and ribbon synapse buys us.

Reproducibility: Yes

Additional Feedback: While a rigorous interpretation of the analysis presented in this work needs additional work, expanding on some of the control analyses suggested above (permitting space limitations) would be useful. ------------------------------- UPDATE: The authors have addressed many concerns, particularly with the ablation study -- please include it in the main paper.


Review 2

Summary and Contributions: The authors present a network model of the retina with biologically realistic constraints including cell types, connectivity patterns (local vs. global), and synaptic vesicle depletion/refilling dynamics. Importantly, the model is still simple enough to be fit to real data. The authors correlate model components with cell type data, including connectivity (from EM) and physiology.

Strengths: The model provides a framework for combining complex, multifaceted data and theory in studying the retina in an intuitive, interpretable way. This builds significantly upon existing neural network models of retinal response that have weak (or no) constraints on the biological plausibility of the network structure and dynamics. The modeling framework is novel and interesting to the NeurIPS community, and it has clear applicability in visual neuroscience.

Weaknesses: One point of concern is that the generalization performance highlighted in table 1 appears on the low side and does not consistently show that strong of a separation between the model classes. Is the training set sufficient to fit the model? Although the authors emphasize synaptic dynamics are not modeled explicitly by LN models as a motivating factor for their choice in including adaptive synapses, it would strengthen the argument to show that the process improves model accuracy. That is, how well would the BCN model do with all of its nonlinear architecture, but without adaptive synapses? Also, what particular adaptive response properties does the dynamic synapse help explain? (side note: at line 104, this synaptic adaptation is only one part of sensitivity adaption. Adaptation has many meanings and computational components ranging from synapses to network-level processing. One should be careful and not overstate the scope of a specific component chosen include to account for temporal adaptation or lack thereof in other models without more specifics.) UPDATE: The addition of an ablation experiment was really helpful here to answer part of my concerns. Plus, that type of test is also helpful for showing experimentalists how this type of model can aide in designing/interpreting experiments. The authors also replied that the natural movies dataset contained more noise because it had fewer trials over which to average. Requiring trial-averaged data for training is an important limitation of this approach in it's current form, especially if one cares about trial-to-trial variability (e.g., Cafaro & Rieke, 2010), and I welcome the additions to discuss this that the authors mentioned in their reply.

Correctness: The claims and methodology are sound. One minor statistical point: The correlation on line 223 for the data in fig 6A reports an extremely small p value (which may be too small if considering any uncertainty/significant figures of the global transient index). This may be due to correlating the GTI and the RRP capacity directly, but because the RRP capacity is shown in a log scale, it may be more accurate to show the rho (and a confidence interval) when correlating GTI with the log RRP.

Clarity: The methodology is presented clearly. One minor exception was how the particular variant of the synaptic release and refill model worked in this model (lines 108-115). The temporal dynamics and model parameters could be more clearly described here.

Relation to Prior Work: The authors motivate their model choices clearly in relation to both statistical and biophysical approaches.

Reproducibility: Yes

Additional Feedback:


Review 3

Summary and Contributions: The authors develop a new model for system identification of bipolar cell responses to synthetic and natural stimuli. Their trained model explains a variety of effects that align with our current understanding of BC circuits. --- Update --- This is a great paper. I still have reservations about the datasets used by the authors, but overall, this is a fantastic approach for modeling neural circuits.

Strengths: This is an exceptionally cool approach, and a fantastic example of how to bridge recent advances in systems neuroscience with neural networks. I think this approach will stand as a template for the field, and raises fascinating questions about the trade-off between model fits and parameter interpretability.

Weaknesses: My main concern is the extent to which the super interesting byproducts of system identification in a biologically constrained model -- in silico effects, connectivity, and parameter interpretability -- are byproducts of the model's design. That is, are these phenomena the result of fitting neural data, and matching model parameters to retinal circuits, or are they caused by the design of the model. I'd really like to see hypothesis testing vs. untrained, random initializations of the BCN to ensure that this isn't the case. To clarify, I'm not voting reject if the random init model shows (e.g.) the same connectivity matrix as the trained model, but I would like to know how emergent these effects are, and I don't think that was adequately assessed. -- Modeling Why are the correlations for "sine" so low? Qualitatively it seems like you are capturing some low-frequency oscilations, but missing the high frequency ones. Perhaps correlation is the wrong metric? Along those lines, qualitatively it looks like LSTM generalizes to these stimuli better than the BCN... -- Connectomics The authors explain that the connectivity matrix they extract from the Helmstaedter dataset is dominated by On and Off SACs + rod BCs. They normalize the matrix to adjust for this finding. Perhaps the imbalance in synaptic connections described by the authors is due to this contact area rule that they (and Helmstaedter) use to identify synapses? This is a decent rule of thumb but far from perfect (as the authors indeed note). In other words, if its the case that the normalization was meant to correct for noisy synapse classification, I'm not sure if it's totally justified.

Correctness: Yes. I noted some slight objections to the connectivity analysis in the Weaknesses (which might be due to me misunderstanding something), but otherwise this is very rigorous work.

Clarity: Yes, this paper is exceptionally well written.

Relation to Prior Work: Yes, this work uses stronger anatomical and biophysical constraints than prior system identification work uses neural networks in retina.

Reproducibility: Yes

Additional Feedback: Is there any way to find the same phenomenon shown for the BCN in the LSTM? The BCN is a nice step towards merging biophysical models with backprop and gradient descent training routines, but I think it is still taking advantage of a large amount of retina-specific knowledge. Perhaps there's a way to identify the ersatz representation of bipolar and amacrine cells within the LSTM? I understand there's no cell types engineered into this model, but perhaps guided by knowledge of its specific computations (i.e., gates could act as a stand in for ACs), or a clustering analysis of its unit responses, you could identify a correspondence between its parameters and those in the neural data? Demonstrating that the LSTM representations are too idiosyncratic or entangled to do such a thing would strengthen the argument in the discussion that "...such predictions would not be easily possible from a pure systems identification approach." -- Modeling Did you ever try reducing the LSTM dimensionality even further? What's the minimal LSTM model that could fit the chirps? Is it possible that one of the LSTMs gates could capture the amacrine cell connectivity, and its cell weights could capture the bipolar cell connectivity? How does the BCN model perform when you drop the sparsity constraint on its weights? -- Connectomics What does the BCN model connectivity matrix look like before training? How much does the training procedure tune model weights to look like the EM connectivity data? How reliable is the similarity between the model and EM data? If you were to retrain the model would you find the same pattern of connections? Or if you were to train on natural images, do you find/expect to find even better correlations with the EM data? -- Comments If you need space you can move the loss function to the appendix. Please move Table 1 up near Figure 2.


Review 4

Summary and Contributions: The manuscript address a computational model of temporal processing in the inner retina, including inhibitory feedback circuits and realistic synaptic release mechanisms. The model based on BC response is seen to generalize to new stimuli including natural movie sequences, performing on par with benchmark black-box models. In pharmacology experiments, the model replicated in silico effects of blocking amacrine cell populations, indicating that it had learned circuit functions. Also, in depth comparisons showed that connectivity patterns learned by the model werel matched to connectivity patterns extracted from connectomics data.

Strengths: The model herein works to provide information about a complex biologic system using biophysical modeling and ML. It succeeds in this effort and is therefore useful on theoretical grounds to a large audience in both CS and computational biology. the authors believe their modeling can be applied elsewhere to other nervous system substructures utilizing in silico testing/experimentation as a validation. There is high novelty in this approach and potential broad implication.

Weaknesses: There are general weaknesses to all biophysical modeling schemes and although difficult, one cannot be sure with such complex systems that all modeling truly resembles the physical/biologic system of said model. Predictions made from the model and informed by experimentation (which is itself in this case a kind of modeling) may not represent what happens in vitro or in vivo.

Correctness: The claims and methods are sound, the methodology as far as I can follow and having run their scripts, is beyond reproach to me in the timetable given to analyze it.

Clarity: The paper is well formatted and written (see next comment).

Relation to Prior Work: The authors do a nice job of comparing their work to the prior field work and establish their work in connection to two large field sub-domains. This was helpful to me as a reader (and was not so over-simplified so as to be inaccurate). Sometimes one can be reading a manuscript and have no context for the utility of what otherwise might be an elegant model or experiment.

Reproducibility: Yes

Additional Feedback:

[Author Response · NeurIPS 2020]

We thank the reviewers for their positive and insightful feedback as well as the research ideas for future work (e.g. the LSTM experiments suggested by R4). All minor comments will be addressed in the revised paper. Here, we briefly reply to selected major points raised by the reviewers (references refer to the main paper):

**Ablation study (R1 & R2)** We further investigated the contribution of the different model parts by training two ablated models, one without AC feedback terms (a deterministic version of the LNR model in [24]) and one without release block but all AC feedback structure.

The ablated models all showed lower training correlation than the BCN model. In particular, they failed to capture certain features of the chirp response: For the LNR model, we found missing "feedback features" (like higher baseline in some On cells or missing responses to small amplitude On steps, Fig. 1A). For the BCN without release block, we found a mismatch in the response to the long On/Off phase of the stimulus as the adaptation processes could not be fully captured (Fig. 1B). Interestingly, the

Figure 1: **Predictions for ablated models on the chirp stimulus. A.** BCN without feedback: prediction for global BC1 (A.1) and global BCR (A.2) response. **B.** BCN without release block: prediction for local BC4 (B.1) and local BC6 (B.2) response. Stimulus is shown in black.

generalization performance of the ablated models was surprisingly good. In particular, the BCN model w/o release block showed high correlation on the natural movie dataset. We found that some of the functional properties of the release block can likely be captured by the feedback structure. Also, the natural movie data does not contain extended light steps, for which the special release properties of the ribbon synapse are so prominent (Fig. 1). We will add a discussion of these additional results to the paper.

**Training vs model structure (R1 & R4)** Comparing the influence of training and model structure is non trivial. As a first step, we ran the best BCN model, kept the stimulus filter and release block fixed but used randomly initialized feedback connectivity weights. The randomly initialized model performed poorly on the training data, but - depending on the strength of the feedback - does surprisingly well on the test data. The evaluation procedure used in the paper (we assign traces from the test data to the output channel of the model with the highest correlation), surely produces an upwards bias. Nevertheless, it seems that for some test conditions, already some unspecific feedback is sufficient, and the model structure contributes strongly to its overall performance. Please also note that in Fig. 4B,C, we are showing the Tonic Release Index (release under baseline), which is not simply explainable by a decrease in inhibition under the drug conditions.

**Data (R1, R2 & R4)** The training data are averaged over many animals/ROIs/repetitions, while the natural movie dataset consists of averages over only five repetitions and the sine dataset of single trial traces, making the two latter substantially more noisy. Furthermore, the datasets were collected under different experimental conditions, making it a harder generalization task because of the domain shift. We will stress this in the revised paper.

**Model inference (R1)** Fitting our model is a non-convex optimization problem. We tried to address this in Section 4.3-4.4 by using the 20 top performing models, for which we found consistent results. Exploring the whole parameter space would need different approaches (like posterior estimation) and is beyond the scope of this manuscript.

**Connectomics (R1 & R4)** For Section 4.3, we already compared our results to randomly sampled weight matrices. Fig. 9 shows a quantification of this comparison, and we will show a randomly generated, sorted matrix for illustration in the revised manuscript. The normalization of the rod BC and SAC connections was done because they can likely not be learned from a simple 1D stimulus, as they serve very specialized functions. This is an interesting direction for future research.

**Frequency analysis (R4)** We additionally analysed whether the BCN and the LSTM capture similar frequency ranges in the responses, using coherence on the generalization data as a measure. We did not find major systematic differences between the two models.

[Meta-Review · NeurIPS 2020]

Four knowledgeable referees support acceptance for the contributions, notably the development of a new model for system identification of bipolar cell responses to synthetic and natural stimuli which explains a variety of effects that align with our current understanding of BC circuits. I also recommend acceptance. However, please consider revising – to include the ablation study suggested by two of the reviewers. Please note that there was some confusion among the reviewers regarding your rebuttal. Why did you plot different cells for (A) and (B)? You also mentioned that the “generalization performance of the ablated models was surprisingly good”. Did this refer to chirps?